# SARS-CoV-2 Seroprevalence in Those Utilizing Public Transportation or Working in the Transportation Industry: A Rapid Review

**DOI:** 10.3390/ijerph191811629

**Published:** 2022-09-15

**Authors:** Aliisa Heiskanen, Yannick Galipeau, Marc-André Langlois, Julian Little, Curtis L. Cooper

**Affiliations:** 1School of Epidemiology and Public Health, Faculty of Medicine, University of Ottawa, Ottawa, ON K1N 6N5, Canada; 2Department of Biochemistry, Microbiology & Immunology, Faculty of Medicine, University of Ottawa, Ottawa, ON K1N 6N5, Canada; 3Centre for Infection, Immunity and Inflammation (CI3), University of Ottawa, Ottawa, ON K1N 6N5, Canada; 4Ottawa Hospital Research Institute, Ottawa, ON K1H 8L6, Canada

**Keywords:** seroprevalence, SARS-CoV-2, COVID-19, public transit, transportation

## Abstract

Proximity and duration of social contact while working or using public transportation may increase users’ risk of SARS-CoV-2 exposure. This review aims to assess evidence of an association between use of public transportation or work in the transportation industry and prevalence of SARS-CoV-2 antibodies as well as to identify factors associated with seropositivity in transit users. A literature search of major databases was conducted from December 2019 to January 2022 using key worlds including “seroprevalence”, “SARS-CoV-2”, and “public transit”. A narrative review of included studies was completed for the following categories: those working in the transportation industry, healthcare workers relying on public transit, and population-based studies. The association between work in the transit industry and seroprevalence varied based on location, demographic characteristics, and test sensitivities. No association was found in healthcare workers. Several population-based studies indicated higher seroprevalence in those using public transit. Overall seroprevalence estimates varied based on geographic location, population demographics, study methodologies, and calendar date of assessment. However, seropositivity was consistently higher in racial minorities and low-income communities.

## 1. Introduction

The coronavirus disease (COVID-19) caused by severe acute respiratory syndrome coronavirus 2 (SARS-CoV-2) remains an international threat with over 550 million confirmed cases and 6 million deaths worldwide [1]. The virus has had a major impact on the transportation industry, shutting down international borders and influencing travel within municipalities. The drastic reduction in public transit ridership is a result of the introduction of remote work, switch to active transportation (i.e., walking or cycling), increased perception of risk associated with public transit use, and government and/or public health recommendations. However, public transit is an essential service with economic benefit that has been continuously relied on by many throughout the pandemic. The increased proximity and duration of social contact accompanying transit use may increase users’ risk of SARS-CoV-2 infection. Racial minorities and those with lower education and income are at particular risk as they are over-represented in those continuing to use public transit during the pandemic and are already at increased risk for SARS-CoV-2 occupational exposure [2,3,4].

Serological testing provides a robust estimate of prior infection by capturing both symptomatic and asymptomatic individuals. This can provide information on virus transmission patterns, measure herd immunity, and predict the number of people susceptible to the virus. With the introduction of vaccinations and evolving variants, serosurveys can help identify gaps in immunity associated with demographic characteristics or risk factors and can focus vaccination programs on susceptible communities. Several systematic reviews have attempted to consolidate and analyze results from hundreds of serosurveys world-wide to determine overall SARS-CoV-2 seroprevalence [5,6,7]. Many serosurveys report the prevalence of SARS-CoV-2 antibodies stratified by sex, gender, health status, and employment history with substantial heterogeneity of results. A systematic review and meta-analysis of 60 countries estimated that the pooled seroprevalence of SARS-CoV-2 was 9.47% as of 30 March 2021 [8]. Several seroprevalence studies have reported similar levels of exposure [9,10,11]; however, results varied by population. This indicates that at the time of study, most of the population had not been exposed to SARS-CoV-2. Seroprevalence is generally found to be similar among men and women [11,12]; however, some studies have found differences between the sexes [8,10]. A literature review of seroprevalence studies found that HCW are often considered a high-risk group; however, those that use adequate PPE are at no higher risk of infection than other groups [12]. Public transportation has been recognized as a high-risk environment for COVID-19 infection prompting the introduction of safety measures such as mask wearing, enhanced cleaning, and reduced capacity to promote physical distancing. Despite this, the risk of infection while using public transport is largely unknown with few studies focusing on this topic and even fewer that evaluate the risk of infection considering social determinants of health such as income and ethnicity. Our literature review aims to assess the prevalence of SARS-CoV-2 seropositivity associated with (1) employment in the transportation industry (e.g., pilots, bus drivers, or taxi drivers) and (2) use of public transportation. Secondary outcomes included identifying demographic characteristics and risk factors associated with seropositivity in these populations.

## 2. Materials and Methods

A literature search of major databases including MEDLINE, Embase, PubMed, and Google Scholar, as well as the MEDRXIV and BIORXIV pre-print servers was performed. Databases were searched from December 2019 to January 2022. Studies were excluded if they were conducted when Omicron became the dominant variant in the Fall of 2021. Only studies that were written in English and incorporated demographic information were included. Populations of interest included those using public transportation and those working in occupations with high risk of exposure to SARS-CoV-2 (i.e., healthcare workers, taxi drivers, or bus operators). The search strategy included a combination of key words such as: “COVID-19” or “SARS-CoV-2”, “respiratory virus”, “seroprevalence”, “public transit”, “public transportation”, “transportation”, “demographic”, “COVID-19 antibodies”, and “socio-demographic”. Citation and reference tracking was performed and relevant materials manually searched. Databases were systematically reviewed. A risk of bias or quality assessment tool was not used; however, studies were informally assessed for limitations or aspects of the study that may have influenced results which were included in the narrative review. Data was extracted using Excel. Studies were organized in accordance with the Human Development Index (HDI) developed by United Nations [13]. This measure is a summary of average achievement in key dimensions such as length and quality of life, knowledge, and standard of living. On this scale, any country scoring higher than 0.8 in 2019 was considered to have very high human development, between 0.7 and 0.799 high human development, 0.55–0.699 medium human development, and less than 0.55 low human development. Outcomes of interest included the effect measure (relative risk and mean difference) of the association between seroprevalence of antibodies to SARS-CoV-2 and any form of transit use and variables that were predictive of seropositivity.

## 3. Results

Twenty-two relevant studies were identified and narratively reviewed. Publications were identified from African, European, North American, South American, and Asian countries. A list of included studies and their characteristics can be found in Table 1. A summary of findings from studies addressing the association between SARS-CoV-2 seropositivity and work in the transportation industry is presented in Table 2. Studies evaluating the association between SARS-CoV-2 seropositivity and use of public transportation were divided by population. Studies concentrating on healthcare workers and the risk associated with their commute to work are presented in Table 3. Those that focus on the risk associated with transit use in the general population are presented in Table 4.

Outcomes included point seroprevalence, prevalence ratios, odds ratios, relative risks, and associated confidence intervals. Key variables associated with SARS-CoV-2 seropositivity were documented.

## 4. Review

### 4.1. Transportation Industry

Essential workers in the transportation industry typically have a high number of social and professional contacts over a comparably large geographic distance which may increase their risk of SARS-CoV-2 exposure. Several serosurveys have purposely targeted industry workers thought to be at high risk of infection during the first wave of the COVID-19 pandemic [14,15,16,17,18,19,20,21,22,23]. The association between work in the transit industry and seroprevalence is far from clear with conflicting results found across numerous studies.

#### 4.1.1. Very High HDI Countries

A population-based study in New York City during the first wave of the pandemic in Spring 2020 found that seroprevalence of SARS-CoV-2 antibodies was significantly higher in those working in the air (25.0%) and ground transportation (35.0%) industries compared to those who did not work outside the home during a government mandated stay at home order (22.2%) [14]. Of note, test validation was not reported. In contrast, another smaller American population-based study by Feehan et al. found no association [14]. This study population was 63.3% female and 66.9% white, with a small number or participants working in the transportation industry (*n* = 11) leading to broad confidence intervals and uncertainty in results [15]. A multivariate analysis was not included, so the effect of demographic and socioeconomic factors was unknown. In a nation-wide longitudinal seroepidemiological study conducted during the first wave of the COVID-19 pandemic in Spain, Pollan et al. found no difference in seroprevalence between those working in the transportation industry and those working from home [16]. This analysis used sampling weights and post-stratification to adjust for selection probabilities and considered clustering by household and census tract when determining seroprevalence. A large representative sample including children, adolescents, and elderly largely of Spanish nationality were recruited. These subpopulations are unlikely to be employed in the transportation industry and could potentially introduce bias if included in the overall estimate of seroprevalence in transit workers. Both Feehan and Pollan found that the only occupation associated with higher seroprevalence was within healthcare. Two Italian studies targeted workers from a variety of occupational sectors to identify high-risk jobs during the first wave of the pandemic. Both studies had comparable overall seroprevalence at approximately 5% and found that work in the transportation industry was not associated with increased prevalence of antibodies to SARS-CoV-2. However, Airoldi et al. identified clusters of cases that were independent of a single industrial sector indicating that the safety measures initiated by corporations could affect the epidemic situation in each company [17]. PPE use and company policy was not measured in this study so authors were unable to explore this hypothesis. Although the overall sample size was large, the number of participants in each occupation group varied greatly. The sample size was small in several industry sectors resulting in broad confidence intervals which may impact the validity of results. Berselli et al.’s study focused on occupational risks and included mostly participants nominated by their employer, however also included self-referred individuals from whom occupational information was not collected [18]. Reason for self-referral and information on symptoms was not collected. Seroprevalence may have been overestimated if those with symptoms were more likely to participate. This study allowed participants to choose from qualitative tests, quantitative tests, or both. Summary results of each test were available, but results would have been strengthened if participants were randomly assigned a test group or if results were stratified by test. They also switched from testing for IgA and IgG using the EUROIMMUN anti-SARS-CoV-2 test to testing for IgG and IgM using the Roch Elecsys Anti-SARS-CoV-2 test kit for IgG and IgM halfway through the study. The reason for the switch was not indicated and could impact results, especially for IgM which was measured using two separate tests and presented as a single result. The United Arab Emirates has a large community of expatriates living and working in unregulated and often precarious conditions (e.g., construction). Alsuwaldi et al. sampled labor camp workers living in these congregate settings and random households after the first wave in Abu Dhabi, United Arab Emirates [19]. Labour camp workers have different socio-demographic characteristics and were analyzed separately from randomly sampled households representing the general population. They found work in the transportation industry was associated with seropositivity in labor camp workers (OR: 2.7; 95% CI: 1.8–4.0), but not in the general population (OR: 1.5; 95% CI: 0.7–3.2) compared to healthcare workers adjusted for age, sex, region, education, nationality, ethnicity, occupation, and contact with a person diagnosed with COVID-19. This may indicate that seropositivity was associated with labour camp conditions such as congregate living and eating in densely populated areas, and not work in the transportation industry itself.

#### 4.1.2. High HDI Countries

Poustchi et al. and Colmenares-Mejiad et al. both performed studies in countries with high HDI (Iran and Colombia, respectively) and found similar seroprevalences between transportation workers and other high-risk occupations (pharmacy workers, retail, health care workers, and other customer-facing staff) [20,21] indicating no increased risk of infection. Poustchi et al.’s study population was approximately 50% male, mostly between the ages of 30–49 and with no contact with confirmed COVID-19 patients. Serology results from a wide range of assays with differing sensitivities and specificities were reported across studies. Poustchi et al. used Pishtaz Teb SARS-CoV-2 ELISA kits which had not been fully assessed before this study and had the lowest sensitivity (66.9% in their validation study) compared to tests used in other included studies where the sensitivity was in the 90% range. Although they accounted for test performance in a sensitivity analysis, a low sensitivity could increase the number of false negatives and underestimate seroprevalence in this population. Tests with high sensitivity and specificity are required to ensure accurate estimation of seropositivity, especially when COVID-19 prevalence in the population is low. The median age of the study population used by Colmenares et al. was 37.4, 59.9% were female, and most were classified as having a socioeconomic status between 2–4 on a six-point scale. Almost all the studies focusing on the transportation industry were conducted in the first wave of the pandemic. However, Colemenares et al. performed their study during the transition period between the first and second waves in Colombia. They recruited participants over four months, a broad time frame during which transmission patterns and public health recommendations frequently changed. It is possible that the staggered resumption of economic sectors and the effect of time could confound the effect of transportation use on seropositivity in this cross-sectional study.

#### 4.1.3. Medium HDI Countries

In a state-wide survey of over 16,000 individuals in Karnataka, India, the population was divided into groups based on occupation risk [22]. Those that worked as bus conductors and auto drivers had twice the odds of seropositivity compared to low-risk individuals (e.g., those attending outpatient departments for minor ailments) (OR: 2.12; 95% CI: 1.28–3.51). The risk groups in this study were poorly defined and recruited from different populations. The low-risk group included only HCW and patients coming in for periodic care or visits to outpatient departments while the moderate and high-risk groups were chosen from the general population. This makes comparisons between groups challenging and reduces study generalizability.

#### 4.1.4. Low HDI Countries

A Togolese study where almost 50% of the study participants worked in air or road transport found that there was no association between work in the transportation industry and seropositivity [23]. Most participants were male (71.6%), had a post-secondary education (51%), and were of Togolese nationality (98%). The overall seroprevalence in the study of 955 individuals was 0.9% (95% CI: 0.4–1.8). This sample size and low seroprevalence prohibited evaluation of the association with key variables.

#### 4.1.5. Transportation Industry Trends

The majority of studies found that there was no association between work in the transportation industry and SARS-CoV-2 seroprevalence or that the risk of working in this industry was comparable to other occupations considered high risk. Of the studies conducted in countries with very high HDI, seroprevalence was only found to be associated with work in the transportation industry by one study [14] and those working in labour camps [19]. The economic inequality seen by those working in labour camps compared to the general population of Abu Dhabi may make this population more comparable with studies in the medium or low HDI category. Both studies conducted in countries with high HDI found seroprevalence was similar among high-risk occupations, while in countries with medium and low HDI results were inconsistent. The study population and restrictions imposed on economic sectors during the pandemic are key factors that should be considered when evaluating the transportation industry. Historically, the transportation industry has been male dominated which was reflected in several studies [19,23] and may limit generalizability of results. Public safety measures such as lockdowns and travel restrictions varied broadly between regions as the pandemic progressed and likely contributed to the heterogeneity of results. This also related to corporate policies as was noted by Airoldi where clusters of cases were found independent of a single industrial sector. Although some general trends were established, results broadly varied between studies and no overall consensus was reached. This was likely implicated by the various study populations, sample methods, test specifications and timing of included studies.

### 4.2. Healthcare Workers

Healthcare worker (HCW) populations have been a focus of serosurveys due to their high rates of exposure to COVID-19 and the potential threat to patients and other healthcare workers if infected. Several studies have examined the occupational and environmental risks faced by HCW during the first wave of the COVID-19 pandemic, including commute via public transport with mixed results.

#### 4.2.1. Very High HDI Countries

A cross-sectional study in a large group of HCW in Switzerland found a gradient of effect, where the proportion of individuals that tested positive for SARS-CoV-2 antibodies increased with the number of times they used public transit in a week. This study also found that wearing a mask while using public transit reduced the odds of seropositivity by 58.0% (OR: 0.42; 95% CI: 0.2–0.9) [24]. Soffin et al. found no association in one hundred forty-three surgeons and anesthesiologists in a specialty hospital in New York City [25]. The small sample included mostly young, white, educated males with few comorbidities which reflected a lower overall seroprevalence (9.8%) compared to other studies conducted in New York during this period (27% and 23.6%; Venugopal and Pathela [14,26]). These results should be interpreted with caution as they may not be representative of the general population. A study evaluating a more representative population of HCW in New York during the same period identified a higher seroprevalence in those that walked to work compared to those that used private transportation. The study population included those over the age of 20, and had significant representation of Hispanic, Black, Asian, and Caucasian ethnicities. No association with mode of travel to work was identified when controlled for ethnicity, close contact with someone diagnosed or with symptoms of COVID-19, not wearing a face mask in healthcare or community setting, unprotected direct contact with infection secretions or excretions, and symptoms associated with COVID-19 infection (fever, cough, myalgia, ageusia, anosmia, nausea, diarrhea) [26]. This study also thoroughly described risk of exposure in concordance with CDC regulations [27], ensuring precise evaluation of risk. Two similar studies conducted in Japan following the first wave of the pandemic reported overall seroprevalence of less than 1% within their respective hospitals (0.7% Yamamoto; 0.43% Nishida) [28,29]. Although a large proportion of workers relied on public transit to travel to work, no association was found with mode of travel. Of note, seropositivity was lower in HCW compared to overall seropositivity measured in the public.

#### 4.2.2. High HDI Countries

Two studies from high HDI countries were identified; both of which found no significant association between seropositivity and transit use in HCWs. The study populations were similar where at least 60% were female and the majority were between 30–50 years of age. Cruz Arenas et al. performed their study in a private hospital not accepting COVID-19 patients, so it seems likely that the risk of exposure for HCW in this hospital would be lower than other locations accepting COVID-19 patients [30].This evaluation had a relatively small sample size (*n* = 300) that was stratified into ten groups according to occupation thereby reducing statistical power compared to the other studies. De Olivieria et al. sampled primarily HCW working in COVID-19 units in São Paulo, Brazil [31]. However, did not clearly define risk of exposure according to frequency of contact with COVID-19 patients, instead classifying risk as high, medium, or low.

#### 4.2.3. Medium HDI Countries

In contrast to the majority of studies focusing on HCWs, one prospective study conducted in India found seroprevalence was significantly higher in those that relied on public transportation (20.0%) or transportation provided by the hospital (16.9%) compared to private modes of transportation (12.0%). However, multivariate analysis was not conducted [32]. The authors speculated that seropositivity was highest in administrative staff and lowest in physicians due to better adherence to control practices. However, PPE use was measured by voluntary disclosure and not controlled for in a multivariate analysis.

#### 4.2.4. Healthcare Worker Trends

All studies conducted in very high and high HDI countries found there was no association between mode of transport and seroprevalence in HCW, indicating that it is not a risk factor in these populations. It is possible that these regions have implemented effective emergency management procedures and PPE use that has protected employees from exposure. The single study that found an association with mode of transport in HCWs was conducted in India, a medium HDI country. This could be implicated with inefficient public transportation systems and mobility problems that are a product of India’s increasing population density. This may lead to longer or more crowded commutes via public transit with increased risk of infection compared to studies conducted in other countries. There was consensus that seropositivity was higher in those with a household or community contact diagnosed with COVID-19 compared to those that encountered COVID-19 patients in the workplace, suggesting personal protective equipment (PPE) use and hospital policies have been effective in reducing the risk of infection in the workplace [25,27,28,31,32]. These combined findings could indicate HCW are diligently following mask wearing and physical distancing procedures while using public transit. However, this was not documented in most studies. All studies relied on retrospective self-reporting of symptoms associated with COVID-19 and PPE use which could introduce recall or reporting bias if participants do not report or are or unwilling to disclose any breach in protocol associated with PPE use. Similarly, the hospital disposition and number of COVID-19 patients accepted may impact risk of exposure and affect reported seroprevalence between studies. Reliance on voluntary participation within hospitals could introduce response bias if those with higher risk of exposure are the ones volunteering to participate.

### 4.3. Population-Based Studies

Well-designed population-based serosurveys can offer more representative findings and more accurate estimate of overall and subgroup infection risk. Several national or state-wide serosurveys have been conducted at different stages of the pandemic to estimate the extent of infection and identify risk factors among the population, including use of public transit.

#### 4.3.1. Very High HDI Countries

Two population-based studies of representative samples were conducted in the United States between May and July 2020, one in Rhode Island [33], and another in Connecticut [34]. Chan et al. found higher seroprevalence in those that relied on public transit (6.0%) compared to those that used a private vehicle (1.9%) and those that walked/biked (2.8%) [33]. Mahajan found no association between transit use and seropositivity in the general population. Of note, public transit use was low (3.4%). However, a greater proportion used public transit in a subgroup of non-Hispanic black and Hispanic participants and the seroprevalence was considerably higher (23.7%) in black transit users [34]. Non-response bias may threaten the validity of results given response rates as low as 11% and 7% in each study, respectively. Chan found response rates lower in men, those under the age of 35, and non-whites. Mahajan did not perform an analysis on those that did not respond but did consider the nonresponse rate when weighting the sample. The potential to introduce bias is high if those that are more likely to test seropositive such as racial minorities have higher non-response rates.

#### 4.3.2. High HDI Countries

A provincial-wide serosurvey in Ecuador during the same period determined that the odds of seropositivity associated with transportation use in a bivariate analysis was 1.73 (95% CI: 1.4–2.2) [35]. This study did not consider assay sensitivity/specificity but did adjust for key covariates (age, gender, having enough resources for living, household COVID-19 contact, contact with the flu, number of occupants in house, physical contact with someone with COVID-19), which did not change the results. In this study the seroprevalence was highest in those using public transit (bus/taxi) (18.2% (95% CI: 15.5–21.2)) compared to those who walked (12.5% (95% CI: 9.6–15.5)), used a bicycle/moto/trolley car (11.3% (95% CI: 6.2–19.3)), or private car (11.0% (95% CI: 9.0–13.1)).

#### 4.3.3. Medium HDI Countries

A national scale study of over 10,000 participants assessed the spread of infection in India from August to September 2020 [36]. Results revealed that those using public transport were almost twice as likely to have antibodies to SARS-CoV-2 compared to those using private transport (OR 1.79; 95% CI: 1.4–2.2). However, the authors did not consider the sensitivity/specificity of the antibody assays utilized or adjust for demographic characteristics which could affect validity of results.

#### 4.3.4. Population-Based Study Trends

The two studies conducted in very high HDI countries had opposing results, likely a result of differing population demographics and test methods. Mahajan was the only study that found transit use was not associated with seropositivity in the general population, but did find SARS-CoV-2 seroprevalence considerably higher in black and Hispanic transit users [28]. Although conducted in two very different locations, the studies conducted in high and medium HDI countries both found that public transit users were almost twice as likely to have antibodies for SARS-CoV-2. This indicates that transit use is likely a risk factor for SARS-CoV-2 infection for the general population regardless of region. Most studies relied on random sampling weighted by census data to obtain a representative sample. It can be difficult to obtain a representative population sample of elderly and young adults using household surveys and census information. Young adults have higher mobility and may be officially registered to live in their parents’ home but have relocated elsewhere. Public health measures and timing of lockdowns within communities may alter the estimated seroprevalence between studies. Acurio-Páez et al. performed their study while restrictions were relaxed and almost all activities were operating normally [35]. They reported over 90% adherence to physical distancing and mask wearing preventative measures, although reporting bias may be prevalent as adherence was measured by voluntary disclosure. In contrast, Chan et al. performed their study during a time of strict lockdown with mandated mask use, where adherence to COVID-19 preventative measures was unknown or not reported [33]. Observing preventative safety measures and adhering to lockdown protocol has been proven to reduce SARS-CoV-2 transmission, and likely the risk of exposure to the public. This could explain why overall seropositivity measured by Acurio-Páez et al. was more than 10% higher than what was found by Chan.

### 4.4. Race and Ethnicity

Race and ethnicity are potential confounding variables when analyzing the relationship between transit use and seropositivity as there are complex interactions between occupational, social, and structural inequalities that increase the risk of COVID-19 in minority communities. Several studies examining socioeconomic factors and spatiotemporal transit use across neighborhoods have shown that decline in magnitude of ridership seen throughout the COVID-19 pandemic is much lower in people of color, those with less education, and lower income [2,3,4]. Black and Hispanic Americans make up a large proportion of the essential workforce, in jobs that require travel and public interaction (i.e., service industry, transportation, health care, cleaners, or seasonal workers) [37]. They are also less likely to own personal cars and require access to public transportation to get to school, work and access essential services such as childcare throughout the COVID-19 pandemic [38]. Several of the studies were conducted in a local population with low diversity [23,35] or did not report information on race or ethnicity [20,21,22,30,31,32,36]. Pollan et al. and Alsuwaldi et al. found that seroprevalence was lower in those of native nationality compared to other nationalities [16,19]. Five of the six studies conducted in the United States sampled a diverse population and found that non-Hispanic black and Hispanic individuals had higher seroprevalence for SARS-CoV-2 antibodies compared to white individuals [14,15,26,33,34]. Venugopal et al. and Pathela et al. both conducted studies during the first wave of the pandemic in New York City and concluded that black and Hispanic individuals were almost twice as likely to have evidence of infection compared to white participants [14,26]. Pathela et al. estimated that the seropositivity in Hispanics was 35.3% (95% CI: 34.4–36.2) and 33.5% (95% CI: 32.0–35.1) in non-Hispanic black compared to non-Hispanic white 16.0% (95% CI: 15.5–16.6). In a regression analysis adjusted for sex, age, area of residence and level of poverty, the risk of seropositivity was almost twice as high in black and Hispanic individuals compared to non-Hispanic white individuals (RR: 1.83; 95% CI: 1.7–1.9 and RR: 1.84; 95% CI: 1.8–1.9), respectively [14]. Very similar results were found by Venugopal et al., where Hispanic and black individuals were more than twice as likely to test positive for SARS-CoV-2 antibodies compared to white participants in unadjusted OR of 2.42 (95% CI: 1.3–4.5) for Hispanics and 2.55 (95% CI: 1.3–5.0) for black participants [26]. These disparities disappear when added to a multivariate analysis that adjusts for ethnicity, mode of travel, moderate or high risk of community or occupational exposure, and symptoms attributable to COVID-19 infection.

### 4.5. Factors Associated with Seropositivity

Overall seroprevalence for SARS-CoV-2 antibodies varied geographically within single studies, where sampling techniques and testing procedures remained consistent. This indicates that regardless of methodology, risk of exposure, demographic variables, and population density likely contribute to risk of SARS-CoV-2 infection. The effect of population density was measured in several studies and revealed a clear relationship between increasing number of people in a region or household and seropositivity for SARS-CoV-2 antibodies. Similarly, higher seroprevalence was found in urban settings where quarantine and separation from household members may not be realistic. An association between flu-like symptoms and SARS-CoV-2 antibody prevalence was consistent across studies. Common symptoms included ageusia, anosmia, fever, diarrhea and olfactory alterations. Comparable to results found in more extensive systematic reviews [7,39], most studies found no association between sex and seropositivity for SARS-CoV-2 antibodies.

Inherent differences between countries with high and low HDI relate to how citizens are impacted by COVID-19 and contribute to the heterologous results of this evaluation. In countries with lower HDI, inadequate housing may make physical distancing and quarantine impractical. There is a large amount of informal economic activity where workers are paid on a daily wage, are self-employed or are involved in informal organizations [40]. A greater proportion of these workers have no choice but to work outside the home during the pandemic, often in hazardous conditions which could lead to greater seroprevalence for SARS-CoV-2. Not all reviewed studies conducted in lower HDI countries reported higher seroprevalence. This could indicate lack of testing capacity or insufficient sampling techniques caused by lack of organization in informal industries. Countries with higher HDI generally experienced higher burdens of COVID-19 earlier in the pandemic [41]. These countries have greater contributions to the global economy encouraging business and tourism travellers that potentiate the spread of infectious disease. This was especially true early in the pandemic prior to the introduction of travel restrictions. Similarly, developed urban areas have widespread and efficient domestic transport systems that support connectivity but also facilitate COVID-19 transmission. Compared to lower HDI countries, higher HDI countries have a higher proportion of elderly people, putting them at greater risk of severe outcomes and mortality [42]. In Asian countries, masks are often worn during influenza season. This may have translated to greater adherence to COVID-19 public safety measures such as mask wearing during public transit use compared to other countries where mask wearing is a novelty. This might explain why COVID-19 seroprevalence was lower in Japanese studies.

## 5. Study Limitations

Several common limitations were observed across included studies. Most were observational cross-sectional studies. As such, they cannot be used to establish temporal association or causality. Cohort or longitudinal studies would be beneficial to analyze the change in seroprevalence and detect seroconversion that could be missed due to early testing or delayed antibody response. Longitudinal studies in particular are required to measure the effect of vaccination rates and introduction of new variants that were not captured in any of the included studies. Many of the studies relied on voluntary participation for recruitment. This method has the potential to introduce bias if those with COVID-19 symptoms are more likely to participate. This concern was evaluated by Pathela et al. who determined that 60% of participants had previous symptoms of COVID-19. Almost all studies relied on self-reported questionnaires to determine transit use and demographic characteristics. This method is prone to self-report bias as participants may provide inaccurate responses to preserve social desirability or control perception of themselves. Alternatively, if questions are ambiguous or wording is not inclusive to all reading levels, participants may interpret questions differently leading to inconsistencies in results. The validity of serosurveys has been questioned due to inconsistent sampling techniques and assay accuracy. High-quality serosurveys use validated tests, standardized lab methods and correct for demographic characteristics and test performance in seroprevalence estimates [43]. Several of the studies included in this analysis did not define the cut-off value used to estimate positivity, which could impact the calculated specificity and result in inaccurate seroprevalence estimates [44]. Similarly, an assortment of serological tests with their own sensitivities and specificities were used to estimate seroprevalence making comparison between studies difficult. Furthermore, the lack of reference samples in the early days of the pandemic complicated assay calibration and standardization. Serosurveys may not capture all past infections due to variations in antibody kinetics such as time to seroconversion, duration of antibody response and rate of antibody waning. Cross-reactivity with common human coronavirus strains and other respiratory viruses is also a concern based on the assay and protein used and could impact seroprevalence results [45,46,47].

This rapid review has several limitations. Most included studies were conducted during the first wave of the pandemic. There are several reasons for this. Many seroprevalence studies were conducted at the beginning of the pandemic to better understand risk factors for this novel disease. Studies were also limited to only include those conducted before the emergence of the Omicron variant. Omicron has vastly different epidemiological characteristics and COVID-19 restrictions have changed with time, so these studies were excluded to improve comparability of results. Another limitation was the presence of several confounding factors that influenced the evaluation of the effect of public transportation use and seroprevalence of SARS-CoV-2. Results of seroprevalence studies are time-dependent and vary based on the location and type of test method. Therefore, conclusions from individual studies reflect the context of each region and may not be applicable to predict seroprevalence in neighboring regions. This limitation reflects the heterogeneity of findings found in this review and other seroprevalence studies worldwide and should be expected as a natural circumstance of the pandemic. Although there are limitations, results from seroprevalence studies do show general trends in levels of population exposure and can help disentangle confounding factors to help identify risk factors as the pandemic evolves.

## 6. Conclusions

Most studies focusing on industry risk factors found no association between SARS-CoV-2 seroprevalence and work in the transportation industry, or that the risk of infection was no higher than other high-risk occupations. However, findings were not universal with conflicting results found in several studies. In HCWs, there was a consensus that mode of transport was not associated with SARS-CoV-2 seroprevalence. In population-based studies, use of public transit was generally associated with higher seropositivity and was implicated by race and ethnicity. A major limitation of these findings was that results could not be generalized due to the involvement of several confounding factors. Of the few studies that considered this topic, results varied based on study population, sociodemographic characteristics, type of assay used, sampling strategies, statistical analysis techniques, occupational hazards and timing of economic closures. However, several studies did report that SARS-CoV-2 seroprevalence was higher in transit users and results from these individual studies should be deliberated. Policy makers should consider transit use a risk factor when planning, implementing, monitoring, and improving strategies for reducing SARS-CoV-2 transmission. In regions where seroprevalence was associated with transit use in transportation workers, employers and policy makers may consider enforcing stricter adherence to public safety measures or implementing additional measures to protect employees. Similarly, across all studies seroprevalence was found to be higher in ethnic minorities and areas with lower socioeconomic status. Policy makers and practitioners should be aware that these populations may have higher rates of exposure and may be more severely impacted by SARS-CoV-2. Further high-quality studies focused solely on this topic are required, as are longitudinal studies as the COVID-19 pandemic progresses.

## Figures and Tables

**Table 1 ijerph-19-11629-t001:** Characteristics of included studies from highest to lowest HDI.

Author	Location	HDI	Date	Type of Study	Study Population	# Participants	Serology			
							Assay	Target	Sensitivity (%) *	Specificity (%) *
Meylan	Lausanne, Switzerland	0.955	18 May–12 June 2020	Cross-sectional	Centre Hospitalier Universitaire Vaudois and Centre for Primary Care and Public Health staff	1874	Luminex-based assay (IgG)	S-protein	97	98
Pathela	New York City, USA	0.926	13 May–21 July 2020	Cross-sectional	NYC adult resident; occupation subgroups	45,367	Liaison SARS-CoV-2 S1/S2	S1/S2 subunits of S protein	97.6	99.3
Soffin	New York City, USA	0.926	6 May–5 June 2020	Cross-sectional	Surgeons and Anaesthesiologists at Hospital for Special Surgery	143	Abbott Architect SARS-CoV-2 IgG	Nucleocapsid	94–100 _a_	99.4–100 _a_
Venugopal	New York City, USA	0.926	May 2020	Cross sectional	Frontline HCWs of NYC hospitals	500	Abbott Architect IgG Assay	Nucleocapsid	100 (95% CI 95.8–100%)	99.6 (95% CI: 99–99.99%)
Feehan	Baton Rouge, USA	0.926	15–31 July 2020	Cross sectional	Representative sample of residents	2138	Abbott Architect i2000SR IgG Assay	Not specified	Not specified	Not specified
Chan	Rhode Island, USA	0.926	5–22 May 2020	Cross-sectional	Households, oversampled African Americans/Blacks and Hispanics/Latinos	1043	Not specified	Not specified	Not specified	Not specified
Mahajan	Connecticut, USA	0.926	4 June–29 July 2020		Adults living in non- congregate settings (exclude those living in LTC homes, nursing homes, prisons); also oversampled non-Hispanic black and Hispanic individuals	567	Ortho-Clinical Diagnostics Vitros anti-SARS-CoV-2 IgG (some negative samples retested with Abbott Architect IgG—targeting nucleocapsid protein)	S-protein	90	100
Yamamoto	Toyama and Kohnoda, Japan	0.919	October–December 2020	Repeated cross-sectional	National Center for Global Health and Medicine employees	2563	Abbott Architect (IgG); Roche Elecsys (total antibodies), confirmatory analysis of positive results using EUROIMMUN anti-S IgG immunoassay	Nucleocapsid protein	Not specified	Not specified
Nishida	Osaka Prefecture, Japan	0.919	12–19 June 2020	Cross-sectional	Toyonaka Municipal Hospital employees	925	Abbott Architect SARS-CoV-2 IgG Assay	Nucleocapsid	100	99.6
Pollan	Spain	0.904	27 April–11 May 2020	Population-based cohort study	Spanish population	66,805	Orient Gene Biotech COVID-19 IgG/IgM Rapid Test Cassette (Point-of-Care Test);	RBD of S protein	IgG: 97.2; IgM: 87.9	100
							Abbott Architect IgG assay	Nucleoprotein	100 _a_	99.6
Airoldi	Piedmont region, Northwest Italy	0.892	28 April–7 August 2020	Cross-sectional	Company workers through screening program	23,568	ZEUS ELISA SARS-CoV-2 IgG Test system	Not specified	93.3 (95% CI: 78.7–98.2)	100 (95% CI: 94.8–100)
Berselli	Emilia Romagna region, Northern Italy	0.892	1 June–25 September 2020	Cross-sectional	Company workers, self-referred individuals	7561	EUROIMMUNE ELISA anti-SARS-CoV-2 test for IgA and IgG	Not specified	100 _c_	92.5
							Roche Elecsys	Not specified	100 _a_	99.8
							KHB SARS-CoV-2 IgM/IgG antibody Colloidal Gold	Not specified	98.81	98.02
Alsuwaldi	Abu, Dhabi, United Arab Emirates	0.890	July 19–August 14 2020	Cross-sectional	Households in region; labour camps	8831 (households); 4855 (labour camp worker)	Roche Elecsys Anti-SARS-CoV-2	Nucleocapsid	100 (95% CI: 88.1–100) _a_	99.8 (95% CI: 99.6–99.1)
							LIAISON SARS-CoV-2 S1/S2 IgG Assay	S1 and S2 subunits of S protein	97.4 (95% CI: 86.6–99.5) _a_	98·5 (95% CI: 97·6–99·1)
Poustchi	18 Iranian Cities	0.783	17 April –2 June 2020	Cross sectional	General population; high-risk occupations	8902	Pishtaz Teb SARS-CoV-2 ELISA IgG and IGM	Not specified	IgG: 94.1; IgM: 79.4	IgG: 98.3; IgM: 97.3
Cruz-Arenas	Mexico City, Mexico	0.779	10 August–9 September 2020	Cross-sectional	Instituto Nacional de Rehabilitación employees	300	LFA: IgG/IgM Rapid Test Cassette;	Not specified	79.5	100
							ELISA: Euroimmun Anti-SARS-CoV-2 NCP IgG Assay	Nucleocapsid protein	Not specified	Not specified
Colmenares-Mejía	Bucaramanga, Colombia	0.767	28 September–24 December 2020	Cross-sectional	Workers from health, construction, public transportation, public force (army, police, transit officers), bike delivery messengers, independent or informal commercial (shopkeepers)	7045	Abbot ARC COV2 (IgG and IgM)	Not specified	85.2	97.3
De Oliveira	São Paulo, Brazil	0.765	March–July 2020	Cross-sectional	Sírio-Libanês Hospital employees	1996	ELISA (IgG), unspecified	Nucleocapsid	86–95 _a_	100 _a_
Acurio-Paez	Cuenca, Ecuador	0.759	11 August–1 November 2020	Cross sectional	Randomly selected inhabitants of Cuenca, Ecuador	2457	SD BIOSNSOR Standard Q COVID-19 IgG/IgM Plus	Not specified	94.3 _b_	87.9 _b_
Babu	Karnataka, India	0.645	3–16 September 2020	Cross-sectional	Statewide population; risk subgroups	16,416	COVID Kavach Anti SARS-CoV-2 IgG antibody detection ELISA	Not specified	92.1	97.7
Gupta	New Delhi, India	0.645	22 June–24 July 2020	Cross-sectional	HCW—All India Institute of Medical Sciences Staff	3739	ADVIA Centaur COV2T chemiluminescence IgG and IgM immunoassay	S-protein RBD	100 _a_	99.8 _a_
Naushin	India	0.645	August–September 2020	Longitudinal, Cohort	Phenome-India Cohort	10,427	Roche Elecsys Anti-SARS-CoV-2; positive samples tested using GENScript cPass SARS-CoV-2 Neutralization Antibody Detection Kit	Nucleocapsid; S-protein	Undefined	Undefined
Halatoko	Lome, Togo	0.515	23 April 2020–8 May 2020	Cross sectional	Occupational sectors: health care, air transport, police, road transport, informal (market sellers, craftsmen)	955	Lungene Rapid Test (IgG and IgM)	Not specified	72.9	85.0

S-protein: spike protein; RBD: receptor binding domain; ELISA: enzyme-linked immunosorbent assay; LFA: lateral flow assay. * Does not include sensitivities and specificities from validation tests performed by authors. _a_ at least 14 days after symptom onset or positive RT-PCR test. _b_ at least 15 days post infection. _c_ at least 10 days after symptom onset.

**Table 2 ijerph-19-11629-t002:** Summary of findings from studies that assessed the association between seropositivity and work in the transportation industry from highest to lowest HDI.

Author	HDI Category	Outcome	Overall Seroprevalence (%)	Transit Outcomes		Variables Associated with Seropositivity	Conclusion
				Seroprevalence (%)	Regression Analysis (i.e., OR, RR)		
Pathela	Very high	Seroprevalence (%), Poisson regression (RR; 95% CI)	23.6% (95% CI: 23.2–24)	Air transport (*n* = 137): 25%; Public transit, taxis and private drivers (*n* = 479): 35%; Other transportation and warehousing (*n* = 440) 27%	Essential worker (food services, construction, retail trade, transportation) compared to other industries RR: 1.63 (95% CI: 1.5–1.7); Adjusted for sex at birth, age, borough, poverty level, working outside the home RR: 1.33 (95% CI: 1.3–1.4)	Male sex, age 44–64, non-White race/ethnicity, living in a borough other than Manhattan or Staten Island, living in neighborhoods with high or very high poverty levels, employment in health care or essential worker category, not being unemployed at the time of serosurvey, working outside the home, having contact with someone with COVID-19, COVID-19 symptoms, being overweight or obese, increasing number of household members	Those working in the transportation industry more likely to have SARS-CoV-2 antibodies
Feehan	Very high	Seroprevalence (%), census weighted bivariate analysis (OR)	3.6% (95% CI: 2.8–4.4)	N/A	Working in the transportation industry (*n* = 11) compared to an office OR: 6 (95% CI: 0.1–100)	Single marital status, public-facing job compared to office, healthcare career, black non-Hispanic race/ethnicity, younger than 29 years old	Work in the transportation industry comparable to risk associated with work in an office
Pollan	Very high	Seroprevalence (%) using two assays	POC test: 5% (95% CI: 4.7–5.4); Immunoassay: 4.6% (95% CI: 4.3–5.0)	POC test: (*n* = 800); 5.9% (95% CI: 3.9–8.7); Immunoassay (*n* = 731): 5.8% (3.6–9.2)	N/A	Province, working in healthcare, confirmed COVID-19 case in household or among non-cohabitating family members and friends or among caregivers and cleaning staff or clients, COVID-19 symptoms	Seroprevalence of those working in the transport industry comparable to overall seroprevalence; comparable between tests
Airoldi	Very high	Seroprevalence (%)	4.97% (95% CI: 4.69–5.25)	4.36% (95% CI: 1.95–6.78)	N/A	Geographical location, those working in logistics or weaving factories	Seroprevalence in transportation industry workers comparable to general population
Berselli	Very high	Seroprevalence (%)	4.7% (95% CI: 4.2–5.2)	1%	N/A	Seroprevalence higher in women, older age groups, HCW, dealers and vehicle repair workers, sport sector employees	No evidence of increased seroprevalence
Alsuwaldi -Household population *	Very high	Seroprevalence (%), bivariate model, multiple logistic regression model (OR)	10.4% (95% CI: 9.5–11.4)	20.8% (95% CI: 15.9–26.7)	OR: 1.5 (95% CI: 0.7–3.2) adjusted for age, sex, region, education, nationality, ethnicity, occupation, contact with someone diagnosed with COVID-19	Households: Region, education level, Asian ethnicity, not from UAE, contact with someone with COVID-19, COVID-19 symptoms	No association with transit use in multivariable analysis
Alsuwaldi -Labour camp population *	Very high	Seroprevalence (%), bivariate model, multiple logistic regression model (OR)	68.6% (95% CI: 61.7–74.7)	72.1% (95% CI: 60.4–81.5)	OR: 2.7 (1.8–4.0) adjusted for age, sex, region, education, nationality, ethnicity, occupation, contact with someone diagnosed with COVID-19	Education, non-Arabic ethnicity, occupation, contact with someone with COVID-19, COVID-19 symptoms	Transit use and high-risk occupations associate with seropositivity
Poustchi	High	Seroprevalence (%) adjusted for population weighting and test performance	General population: 17.1% (95% CI: 14.6–19.5); High-risk population: 20% (95% CI: 18.5–21.7)	Taxi drivers (*n* = 718): 18.8% (95% CI: 14.7–23.2)	N/A	60 years or older, those in contact with someone with COVID-19, region, COVID-19 symptoms	Seroprevalence similar between high-risk occupations
Colmenares-Mejía	High	Seroprevalence (%) corrected for test performance and study design	19.5% (95% CI: 18.6–20.4)	Commute to work: bike: 25.7% (95% CI: 16.6–34.8); public transportation: 23.9% (95% CI: 21.8–26); taxi 15.5% (95% CI: 12.3–18.7) Those working in the public transport industry: 16% (95% CI: 11.7–20.3)	N/A	Occupational groups with multiple contacts with others during work hours, delivery drivers, grocery store tenants, informal commerce workers, those that used a bike, motorcycle, public transit than own car, COVID-19 symptoms	Similar seroprevalence in those working in the transportation industry and other high-risk occupations. Higher seroprevalence in those that use public transit to commute to work compared to those that use their own vehicle
Babu	Medium	Seroprevalence (%), generalized linear model-based multinomial regression (OR)	16.8% (95% CI: 15.5–18.1)	Bus conductors/auto drivers (*n* = 1008): 16.1% (95% CI: 11.7–20.6);	Bus conductors/auto drivers compared to low-risk occupations: OR: 2.12 (95% CI: 1.3–3.5)	Diarrhoea, chest-pain, rhinorrhea, fatigue, fever, professions who had more contact with the public, residence in containment zones, urbanisation level of the district	Those working in the transportation industry twice as likely to have SARS-CoV-2 antibodies
Halatoko	Low	Seroprevalence (%)	IgM or IgG: 0.9% (95% CI: 0.4–1.8)	Air transport (*n* = 212); IgM positive: 0.5% (95% CI: 0.01–2.6); IgG positive: 0.9% 95% CI: 0.1–3.4) Road Transport (*n* = 122) IgM positive: 0% (95% CI: 0–2.9); IgG 0.8% 95% CI: 0–4.5)	N/A	N/A	Low seroprevalence in general, similar among high-risk populations

RR: relative risk; OR: odds ratio; POC: point of care. * Alsuwaldi reported separate results for two different populations. Results varied by population so were presented as two separate studies.

**Table 3 ijerph-19-11629-t003:** Summary of findings from studies that assessed the association between seropositivity and use of public transportation in healthcare workers from highest to lowest HDI.

Author	HDI Category	Outcome	Overall Seroprevalence (%)	Transit Outcomes		Variables Associated with Seropositivity	Conclusions
				Seroprevalence (%)	Regression Analysis (i.e., OR, RR)		
Meylan	Very high	Seropositivity (%), multivariable logistic regression (OR)	10% (95% CI: 8.7–11.5)	Frequency of transit use (# per week) 1 (*n* = 104): 7.7% (95% CI: 3.4–14.6); 2 (*n* = 135): 9.6% (95% CI: 5.2–15.9); 3 (*n* = 148): 10.1% (95% CI: 5.8–16.2); 4 (*n* = 199): 12.1 (95% CI: 7.9–17.4); 5 (*n* = 275): 14.2% (95% CI: 10.3–18.9); >5 (*n* = 220): 5.9% (95% CI: 3.2–9.9); Use of face mask on public transport (*n* = 151): 5.3% (95% CI: 2.3–10.2) Does not use face mask on public transport (*n* = 930): 11.2% (95% CI: 9.2–13.4)	Use of mask at public transport compared to those that do not: OR = 0.42 (95% CI: 0.198–0.896) adjusted for daily contact w patients, work in ICU, COVID-19 case at home, and COVID-19 symptoms	Household contact with confirmed COVID-19, use of mask while using public transport, COVID-19 symptoms	Seropositivity increased with transit usage; face mask while using public transit reduced odds of seropositivity
Soffin	Very high	Seroprevalence (%), bivariate logistic regression (OR)	9.8%	N/A	OR: 1.48 (95% CI: 0.2–6.3)	Fatigue, myalgia, fever, headache, spouse diagnosed with COVID-19	No association with mode of commute (public transport, walking/cycling, private)
Venugopal	Very high	Seroprevalence (%), bivariable, multivariable linear regression (OR)	27%	29%	Public transit compared to private OR: 1.3 (95% CI: 0.9–2.0) Adjusted for ethnicity, symptoms, duration of symptoms: OR: 0.84 (95% CI: 0.47–1.52)	Ethnicity other than Caucasian, living in an apartment/condo, walking to work, symptoms of COVID-19, community exposure	Type of transport to hospital not associated with seropositivity
Yamamoto	Very high	Seropositivity (%), Poisson regression (PR)	0.7% (95% CI:0.4–1.1)	N/A	Compared to those that used transit <1 time/week, those that used it 1 or more times/week prevalence ratio was 0.57 (95% CI: 0.2–1.4)	Close contact with patients with COVID-19 at home and in the community	No association with transit
Nishida	Very high	Seropositivity (%)	IgG: 0.43% (95% CI:0.2–1.1)	0.76% (*n* = 396) (95% CI: 0.3–2.2)	N/A	No significant factors	No association with transit
Cruz-Arenas	High	Seropositivity (%), multiple logistic regression (OR)	LFA: 11% ELISA: (IgG only) 13%	N/A	Use of public transport for work commute OR: 1.62 95% CI: 0.82–3.21)	Olfactory alterations, security or janitorial occupations, education below a university degree increasing number of people in household	Type of transport to hospital not associated with seropositivity
De Oliveira	High	Prevalence (%), bivariate analysis, multivariate logistic regression (OR)	5.5%	N/A	Public transport (bus, metro): OR 1.17 (95% CI: 0.79–1.75) Adjusted for gender, cleaning, working at COVID-19 units, type of transport OR: 1.103 (95% CI: 0.731–1.665)	Professional category of cleaning and male gender	Type of transport to hospital not associated with seropositivity
Gupta	Medium	Seroprevalence (%)	13%	Public transit (*n* = 235): 20%; Hospital transport (*n* = 676): 16.9%; Own vehicle (*n* = 1986): 12.4%; on foot (*n* = 544): 11.2%; did not declare (*n* = 298): 6%	N/A	Contact with COVID positive individuals, COVID-19 symptoms, region of residence	Seroprevalence significantly higher in HCW that used public, or hospital transit compared to those that used other modes of commute (*p* < 0.05)

OR: odds ratio; PR: prevalence ratio.

**Table 4 ijerph-19-11629-t004:** Summary of findings from studies that assessed the association between seropositivity and use of public transportation in population-based studies from highest to lowest HDI.

Author	HDI Category	Outcome	Overall Seroprevalence (%)	Transit Outcomes		Variables Associated with Seropositivity	Conclusions
				Seroprevalence (%)	Regression Analysis (i.e., OR, RR)		
Chan	Very high	Seroprevalence (%), age weighted	2.9% (95% CI: 1–6.2)	Public transportation/carpool (*n* = 52): 6% (95% CI: 0.1–20.5); own vehicle (*n* = 920): 1.9% (95% CI: 0.4–4.1); walking/biking (*n* = 34): 2.8% (95% CI: 0–16.7)	N/A	Those living in a condo or apartment, those that rely on public transportation or carpool, race/ethnicity other than Caucasian, primary mode of transportation	Higher seroprevalence in transit users
Mahajan	Very high	Seroprevalence (%), weighted for non-response and population characteristics of Connecticut	General population: 4% (90% CI: 2–6); non-Hispanic black subpopulation: 6.4% (90% CI: 0.9–11.9); Hispanic subpopulation: 19.9% (90% CI: 13.2–26.6)	General population: 0% or too small to calculate Non-Hispanic Black subpopulation Airplane: 4% (±4.8); public transportation: 23.7% (±7.5) Hispanic subpopulation Airplane: 4.8% (±3.3); public transportation: 13.1% (±5.5) *	N/A	Race and ethnicity	No association with transit use in general population, seroprevalence significantly higher in transit users of ethnic minorities
Acurio-Paez	High	Seroprevalence (%), bivariate regression, multivariate regression (OR)	Maximum: 13.2% (95% CI: 12–14.6) (IgG or IgM); Minimum: 4% (95% CI: 3.2–4.8) (IgG and IgM positive)	Foot (*n* = 529): 12.5% (9.6–15.5); Bicycle/moto/trolley car (*n* = 106): 11.3% (6.2–19.3); Private car (*n* = 912): 11% (9.0–13.1); Public (bus/taxi) (*n* = 742): 18.2%(15.5–21.2)	Public (bus/taxi) compared to private (own car, foot, bicycle) OR: 1.73 (95% CI: 1.4–2.2) Adjusted for age, resources, COVID-19 in household, contact with flu-like symptoms, number of people in household, physical contact with someone outside the household: 1.65 (95% CI: 1.28–2.14)	Age 35–49 years old, COVID-19 positive person in the home, using public transit, at least 6 people in a household, physical contact with a person outside the household, contact with someone with flu-like symptoms, not having enough resources for living	Those using public transit at increased risk of seropositivity
Nausin	Medium	Seropositivity (%), bivariate logistic regression (OR)	10.14% (95% CI: 9.6–10.7)	N/A	OR: 1.79 (95% CI: 1.4–2.2) OR males: 1.91 (95% CI: 1.44–2.55); OR females: 1.83 (95% CI: 1.26–2.69)	Higher population density, high exposure work, those using public transit, non-smokers	Those using public transit at increased risk of seropositivity

* ± margin of error at 90% CI.

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
