# Peer review of "SARS-CoV-2 Seroprevalence in Those Utilizing Public Transportation or Working in the Transportation Industry: A Rapid Review"

_ijerph, 2022, doi:10.3390/ijerph191811629_

Round 1
Reviewer 1 Report (Previous Reviewer 2)
The manuscript by Aliisa Heiskanen et al. is in a better shape after re-organized based on human development index. Overall, the included studies in this review article have very mixed conclusions. The authors have very insightful comments on the limitations of these studies. I have a few concerns as below:
1, Since you classify these studies into categories of very high HDI, high HDI, and medium HDI countries. Please have thorough discussion about differences in these 3 categories contributing to the heterologous conclusions.
2, Some of the studies you listed have clear comparisons that are relevant to the major topic, e.g. comparison between pubic transit and private commuting, however, information about comparison subjects in the other studies are very ambiguous.
3, Despite that conclusions from different studies vary, it’s relatively definitive that seropositivity was consistently higher in racial minorities and low-income communities. Did the studies you listed reveal any information about racial make of those working in the transportation industry or healthcare workers in the various studies? If so, please discuss.
Author Response
Point 1: Since you classify these studies into categories of very high HDI, high HDI, and medium HDI countries. Please have thorough discussion about differences in these 3 categories contributing to the heterologous conclusions.
RESPONSE: Thank you for bringing up this important point. To address this comment, a discussion on the differences between HDIs has been added to the factors associated with seropositivity section (Section 4.5, page 25, paragraph 2).
Point 2: Some of the studies you listed have clear comparisons that are relevant to the major topic, e.g. comparison between pubic transit and private commuting, however, information about comparison subjects in the other studies are very ambiguous.
RESPONSE: Thank you for bringing this point to our attention. To address this comment, we have included more information on the included study populations.
Point 3: Despite that conclusions from different studies vary, it’s relatively definitive that seropositivity was consistently higher in racial minorities and low-income communities. Did the studies you listed reveal any information about racial make of those working in the transportation industry or healthcare workers in the various studies? If so, please discuss.
RESPONSE: Where applicable, additional information on race has been added. Unfortunately, most of the studies that targeted workers in the transportation industry or healthcare workers were conducted in populations with low diversity or did not report information on race or ethnicity. For this reason, it was not possible to make inferences about the racial make of those working in the transportation industry or HCW. However, this is an important topic that would be interesting to research in the future.
Reviewer 2 Report (Previous Reviewer 1)
The authors (AA) have carefully addressed the reviewers' comments. Overall the changes made have improved the manuscript, but there are some points to better clarify.
Review:
Lines 126-127: AA should report period and they should insert references (“Several serosurveys have purposely targeted industry workers thought to be at high risk of infection.”)
Lines 149-152: These sentences are not clear.
Line 176: Check it and change IgM with IgG (“[…] IgA and IgM using the EUROIMMUN […]”)
Lines 336 and 341: What period?
Study limitations:
AA should report the presence of several confounding factors for risk evaluation about users of public transportation.
Conclusions:
AA should underline that they cannot generalize their results beacause many confounding factors got involved in the evaluation of the effect of public transportation use on seroprevalence of SARS-CoV-2 antibodies.
Author Response
Point 1: Lines 126-127: AA should report period and they should insert references (“Several serosurveys have purposely targeted industry workers thought to be at high risk of infection.”)
RESPONSE: The report period and references have been added.
Point 2:  Lines 149-152: These sentences are not clear.
RESPONSE: Thank you for bringing this to our attention. These sentences have been changed to improve clarity (lines 155-158).
Point 3: Line 176: Check it and change IgM with IgG (“[…] IgA and IgM using the EUROIMMUN […]”)
RESPONSE: Thank you for catching this error. It has been checked and corrected (line 170).
Point 4: Lines 336 and 341: What period?
RESPONSE: The period has been added “during the first wave of the pandemic” (line 268).
 Point 5: AA should report the presence of several confounding factors for risk evaluation about users of public transportation.
RESPONSE: This is a very important point, we completely agree. The presence of several confounding factors interfering with the evaluation of the effect of public transportation use and seroprevalence of SARS-CoV-2 has been reported in the limitations section (section 5, page 26, lines 537-539).
Conclusions:
 Point 6: AA should underline that they cannot generalize their results because many confounding factors got involved in the evaluation of the effect of public transportation use on seroprevalence of SARS-CoV-2 antibodies.
RESPONSE: The impact of confounding variables in this evaluation has been noted as a limiting factor that prevented generalizability in the conclusion section of this paper (conclusion, page 27, line 558).
Round 2
Reviewer 1 Report (Previous Reviewer 2)
The manuscript compiled the studies related to association between use of public transport and seroprevalence. A total of 22 relevant studies were reviewed and organized based on frequency of using transportation ( commute by public transportation, working in transportation sector, healthcare workers), and Human Development Index (HDI). The limited number of studies and the varying limitations for the studies within the same category defy comparison between different studies. Despite the essence and limitations of these studies, this manuscript did a good review, objectively judging these studies and making insightful comments. I have no further comments on this manuscript.
Reviewer 2 Report (Previous Reviewer 1)
The authors (AA) have carefully addressed the reviewers' comments. Overall the changes made have improved the manuscript.
This manuscript is a resubmission of an earlier submission. The following is a list of the peer review reports and author responses from that submission.
Round 1
Reviewer 1 Report
The authors (AA) aim to assess the prevalence of SARS-CoV-2 seropositivity associated with employment in the transportation industry and use of public transportation. Moreover, AA investigated demographic characteristics and risk factors associated with seropositivity in these populations. This is an article useful to increase our knowledge of the issue. Addressing all the issues below reported could make this manuscript eligible for the publication.
The title could be improved reporting the inclusion of not only users of public transportation but also workers of transportation industry.
Abstract:
AA should report the search period. Moreover, AA should better explain line 66 (is this results about general population or regarding only users of public transportation and/or workers of transportation industry?).
Introduction:
AA should add literature about seroprevalence studies such as:
Palmateer, N.E.; Dickson, E.; Furrie, E.; Godber, I.; Goldberg, D.J.; Gousias, P.; Jarvis, L.; Mathie, L.; Mavin, S.; McMenamin, J.; et al. National population prevalence of antibodies to SARS-CoV-2 in Scotland during the first and second waves of the COVID-19 pandemic. Public Health 2021, 198, 102–105.
Rostami, A.; Sepidarkish, M.; Fazlzadeh, A.; Mokdad, A.H.; Sattarnezhad, A.; Esfandyari, S.; Riahi, S.M.; Mollalo, A.; Dooki, M.E.;Bayani, M.; et al. Update on SARS-CoV-2 seroprevalence: Regional and worldwide. Clin. Microbiol. Infect. 2021, 27, 1762–1771.
Paduano S, Galante P, Berselli N, Ugolotti L, Modenese A, Poggi A, Malavolti M, Turchi S, Marchesi I, Vivoli R, Perlini P, Bellucci R, Gobba F, Vinceti M, Filippini T, Bargellini A. Seroprevalence Survey of Anti-SARS-CoV-2 Antibodies in a Population of Emilia-Romagna Region, Northern Italy. Int J Environ Res Public Health. 2022 Jun 27;19(13):7882.
Kislaya, I.; Goncalves, P.; Gomez, V.; Gaio, V.; Roquette, R.; Barreto, M.; Sousa-Uva, M.; Torres, A.R.; Santos, J.; Matos, R.; et al.SARS-CoV-2 seroprevalence in Portugal following the third epidemic wave: Results of the second National Serological Survey (ISN2COVID-19). Infect. Dis. (Lond) 2022, 54, 418–424.
Lai, C.C.; Wang, J.H.; Hsueh, P.R. Population-based seroprevalence surveys of anti-SARS-CoV-2 antibody: An up-to-date review. Int. J. Infect. Dis 2020, 101, 314–322.
Line 44-54: AA could better explain the gap of knowledge and the novelty of their study.
Methods:
Line 66-69: the search strategy seems a little bit limited with the risk of excluding several relevant papers. Why did AA only include “public transit”? AA should include other terms in their research strategy.
Lines 71-72: What do the authors want to mean when they say that risk of bias assessment and quality was not formally evaluated?
Results / Review:
The presentation of results is a little bit confusing. AA should reorganize the presentation of Results section and Review section. Moreover, there are several studies about seroprevalence in occupational settings including transportation sector, such as:
Airoldi, C.; Calcagno, A.; Di Perri, G.; Valinotto, R.; Gallo, L.; Locana, E.; Trunfio, M.; Patrucco, F.; Vineis, P.; Faggiano, F. Seroprevalence of SARS-CoV-2 among workers in Northern Italy. Ann. Work Expo Health 2021, 66, 224–232.
Berselli, N.; Filippini, T.; Paduano, S.; Malavolti, M.; Modenese, A.; Gobba, F.; Borella, P.; Marchesi, I.; Vivoli, R.; Perlini, P.; et al. Seroprevalence of anti-SARS-CoV-2 antibodies in the Northern Italy population before the COVID-19 second wave. Int. J. Occup. Med. Environ. Health 2022, 35, 63–74.
Modenese, A.; Mazzoli, T.; Berselli, N.; Ferrari, D.; Bargellini, A.; Borella, P.; Filippini, T.; Marchesi, I.; Paduano, S.; Vinceti, M.; et al. Frequency of Anti-SARS-CoV-2 Antibodies in Various Occupational Sectors in an Industrialized Area of Northern Italy from May to October 2020. Int. J. Environ. Res. Public Health 2021, 18, 7948.
Why did the authors decide not to include them?
Conclusions:
Conclusions should be synthetized addressing it to the aims of the study. AA did not report any conclusion about workers of transportation industry.
Reviewer 2 Report
The review article by Aliisa Heiskanen et al. summarized 20 epidemiological studies investigating association between seroprevalence of SARS-CoV-2 antibodies and use of public transit. Generally, this manuscript was well written, gave a comprehensive review of related epidemiological studies. Unfortunately, some limitations of this manuscript made me decide not to recommend this paper to be published in the IJERPH.
1, Many confounding factors got involved in the evaluation of the effect of public transit use on seroprevalence of SARS-CoV-2 antibodies and the reviewed studies had conflict conclusions with one another. Therefore, not too much useful information can be found to improve preventative measures for COVID-19.
2, More importantly, the reviewed studies were all performed at the height of the pandemic when there were strict COVID-19 restrictions, working from home was very common, and no vaccines were available. By comparison, currently, the most prevalent SARS-CoV-2 variant is omicron with very different epidemiological characteristics, a large fraction of the populations has been vaccinated, COVID-19 restrictions are loose, and many more people are commuting to work. Up-to-date epidemiological studies are more relevant to the improvement of preventative measures for COVID-19.